# Reconstituting the complete biosynthesis of D-lysergic acid in yeast

Garrett Wong [1,2,3,4,7], Li Rong Lim [1,2,3,7], Yong Quan Tan[1,2,3], Maybelle Kho Go[1,2,3], David J. Bell [4], Paul S. Freemont [4,5,6✉] & Wen Shan Yew [1,2,3✉]

The ergot alkaloids are a class of natural products known for their pharmacologically privileged molecular structure that are used in the treatment of neurological ailments, such as Parkinsonism and dementia. Their synthesis via chemical and biological routes are therefore of industrial relevance, but suffer from several challenges. Current chemical synthesis methods involve long, multi-step reactions with harsh conditions and are not enantioselective; biological methods utilizing ergot fungi, produce an assortment of products that complicate product recovery, and are susceptible to strain degradation. Reconstituting the ergot alkaloid pathway in a strain strongly amenable for liquid fermentation, could potentially resolve these issues. In this work, we report the production of the main ergoline therapeutic precursor, D-lysergic acid, to a titre of 1.7 mg L$^{-1}$ in a 1 L bioreactor. Our work demonstrates the proof-of-concept for the biological production of ergoline-derived compounds from sugar in an engineered yeast chassis.

¹ Synthetic Biology for Clinical and Technological Innovation, National University of Singapore, 28 Medical Drive, Singapore 117456, Singapore. ² Synthetic Biology Translational Research Programme, Yong Loo Lin School of Medicine, National University of Singapore, 14 Medical Drive, Singapore 117599, Singapore. ³ Department of Biochemistry, Yong Loo Lin School of Medicine, National University of Singapore, 8 Medical Drive, Singapore 117597, Singapore. ⁴ Department of Infectious Diseases, Faculty of Medicine, Imperial College London, Exhibition Road, South Kensington, London SW7 2AZ, UK. ⁵ London Biofoundry, Imperial College Translation & Innovation Hub, White City Campus, 80 Wood Lane, London W12 0BZ, UK. ⁶ UK Dementia Research Institute Care Research and Technology Centre, Imperial College London, Hammersmith Campus, Du Cane Road, London W12 0NN, UK. ⁷These authors contributed equally: Garrett Wong, Li Rong Lim. ✉email: p.freemont@imperial.ac.uk; wenshanyew@nus.edu.sg

The ergot alkaloids have been used extensively as therapeutics throughout history[1,2]. The pharmacological effect of ergot alkaloids have been attributed to the molecular similarity between the ergoline skeleton and the monoamine neurotransmitters, such as adrenaline, dopamine, and serotonin)[3,4]. The ergot alkaloids and their synthetic derivatives are used extensively in modern medicine for the treatment of several neurological ailments, such as Parkinsonism[5], dementia[6], and hypertension[7].

The ergoline pharmacophore is an important scaffold for potential therapeutic discovery and development, particularly in the treatment of neurological and psychiatric disorders. As an illustration, D-lysergic acid diethylamide (LSD), a chemically-derived ergot alkaloid, is one of the most potent agonists for the 5-HT$_{2A}$ serotonergic receptor ($K_d = 0.33$ nM)[8]. The key active pharmaceutical ingredient (API) of these ergoline-derivatives comes from D-lysergic acid (DLA).

To meet the global demand for DLA, 8 tons of ergopeptines and up to 10–15 tons of DLA are produced each year. Approximately 60% of these are produced by the submerged fermentation of specially developed strains of *C. purpurea*, while the rest are obtained from field cultivation[9]. The key limitations to these existing production methods are: firstly the large variation of ergot alkaloids produced which complicates the downstream extraction workflows and drives up the cost of production[10]; and second the tendency for these strains to degenerate over the cultivation and preservation processes[9].

A number of chemical total synthesis routes towards DLA have been reported[11]. However, such routes are highly complex, requiring 8 to 19 chemical transformation steps, and the product is often not enantiomerically pure. For example, the highest yielding process requires 19 steps and produces a reported yield of 12%[12]. In contrast, the simplest method is an 8-step process that produces a reported yield of 10.6% but is not enantioselective[13]. These issues severely impede the use of chemical synthesis to meet the commercial demand for DLA, which is evident in their lack of use in industry.

We propose that the biosynthesis of DLA using an industrially tractable microorganism, such as baker's yeast, would likely circumvent the aforementioned key issues with current methods of production. The use of a well-established host organism would ameliorate any potential strain degeneration, or minimally be simpler to troubleshoot. On the other hand, reconstituting the ergot pathway with the selective enzymes directing the metabolic flux towards the desired branchpoints in a heterologous system, would ensure limited to no variation in the ergot alkaloid profile produced. Here, we show the complete reconstitution of the DLA biosynthetic pathway in baker's yeast, *Saccharomyces cerevisiae*. Our strain, without extensive strain or bioprocess optimization, was able to achieve a titer of 1.7 mg L$^{-1}$ and 1.4 mg L$^{-1}$ in 1 and 4 L scale fermentations, respectively.

## Results and discussion

**Biosynthetic resolution of the ergot alkaloid pathway.** The ergot alkaloids are broadly classified into three groups—the clavines, ergoamides, and the ergopeptines, all of which are distinguished by the different modifications appended to the core ergoline structure. This class of compounds are produced by several filamentous fungi from the *Ascomycota* phylum, but most notably from the parasitic fungus—*Claviceps purpurea*, more commonly known as the ergot fungus and hence their name[14].

The complete biosynthesis of DLA from L-tryptophan requires eight enzymes encoded by the following genes—DmaW, EasF, EasC, EasE, EasD, EasA, EasG, and CloA[10]. The transformations from DmaW to EasD have been biochemically characterized

(Fig. 1)[10,15–18]. All ergot alkaloids are derived from L-tryptophan and share a common set of early biosynthetic steps that forms the ergoline C-ring. The presence of various isoforms of enzymes from different species of ergot producing organisms, involved in the middle and late biosynthetic steps, determine the product profiles produced by these respective organisms[19]. Some of the enzymes have, however, been reported to be refractory toward heterologous expression[17,19]. We therefore sought to find alternative orthologues for these enzymes (encoded by genes *easE*, *easA*, and *cloA*) (Supplementary Fig. 6) to reconstitute the pathway, using the Enzyme Function Initiative-Enzyme Similarity Tool (EFI-EST)[20,21] (Supplementary Discussion 1).

**Screening for the functional expression of *easE* in yeast.** The enzymes EasC and EasE have been reported to both, be essential in the conversion of 4DMA to chanoclavine-I in earlier publications from several groups[17,22–24]. However, EasE from most ergot producing fungi have been shown to have non-optimal activity in heterologous yeast systems[17]. To date, only the EasE orthologue from *Aspergillus japonicus* (EasE_Aj), reported by Nielsen, C.A., et. al. (2014), has been functionally expressed in *S. cerevisiae*[17].

Therefore, to identify additional active EasE orthologues, we used this orthologue to generate a Sequence Similarity Network (SSN). We then identified the putative isofunctional cluster around EasE_Aj and selected eight sequences to screen for expression and enzymatic function (Supplementary Discussion 1). To facilitate this screen, we created a modified strain (YMC17; Supplementary Table 9) with the genes: *dmaW*, *easF*, and *easC*; stably integrated into its genome at the *YMRWδ15* site. The eight EasE orthologues were then transformed into YMC17, on an episomal vector. Out of the eight orthologues, only the orthologues from *Epichloe coenophialia* (EasE_Ec) and EasE_Aj exhibited detectable production of chanoclavine-I (Fig. 2b, Supplementary Fig. 7). Comparisons of the relative amounts of chanoclavine-I produced by EasE_Ec were 10 times lower compared to that from easE_Aj (Fig. 2c). Nevertheless, both sequences could be used to complete the ergot alkaloid pathway in yeast.

**Identifying isomerase variants of *easA*.** The next biosynthetic step in the construction of the DLA pathway involves the branch point linking the tricyclic clavines to the tetracyclic ones. The isoforms of EasA, from different lineages of ergot alkaloid producing organisms, catalyze either a reduction or a cis-trans isomerization across the C8-C9 double bond of the ergoline moiety to position the aldehyde group for EasG to then link it with the methylamino group to form the ergoline D ring[25]. The reduction or retention of the C8-C9 double bond at the end of this process depends on the EasA isoform, diverging the pathway toward either agroclavine, festuclavine, pyroclavine, or if a particular orthologue of EasH is present, cycloclavine (Fig. 3a)[19,25,26].

Directing the metabolic flux towards the agroclavine branch of the pathway, and thereafter DLA, requires the isomerase variant of EasA. As before, we generated an SSN of EasA, with the sequence from *C. purpurea*, to identify an isomerase isofunctional cluster. An earlier publication by Cheng, J.Z., et. al. (2010), reported the importance of the F176 residue, four sequences from this cluster with the structurally corresponding F176 residue were then selected and synthesized[26] (Supplementary Discussion 1). These orthologues were then screened by co-expression with *easD* and *easG* in a strain with *dmaW*, *easF*, *easC*, and *easE_Aj* integrated in the yeast genome (YOCE; Supplementary Table 9).

All *easA* orthologues produced a compound with a [M + H]$^+$ of 239 m/z, corresponding to the agroclavine standard (Fig. 3b).

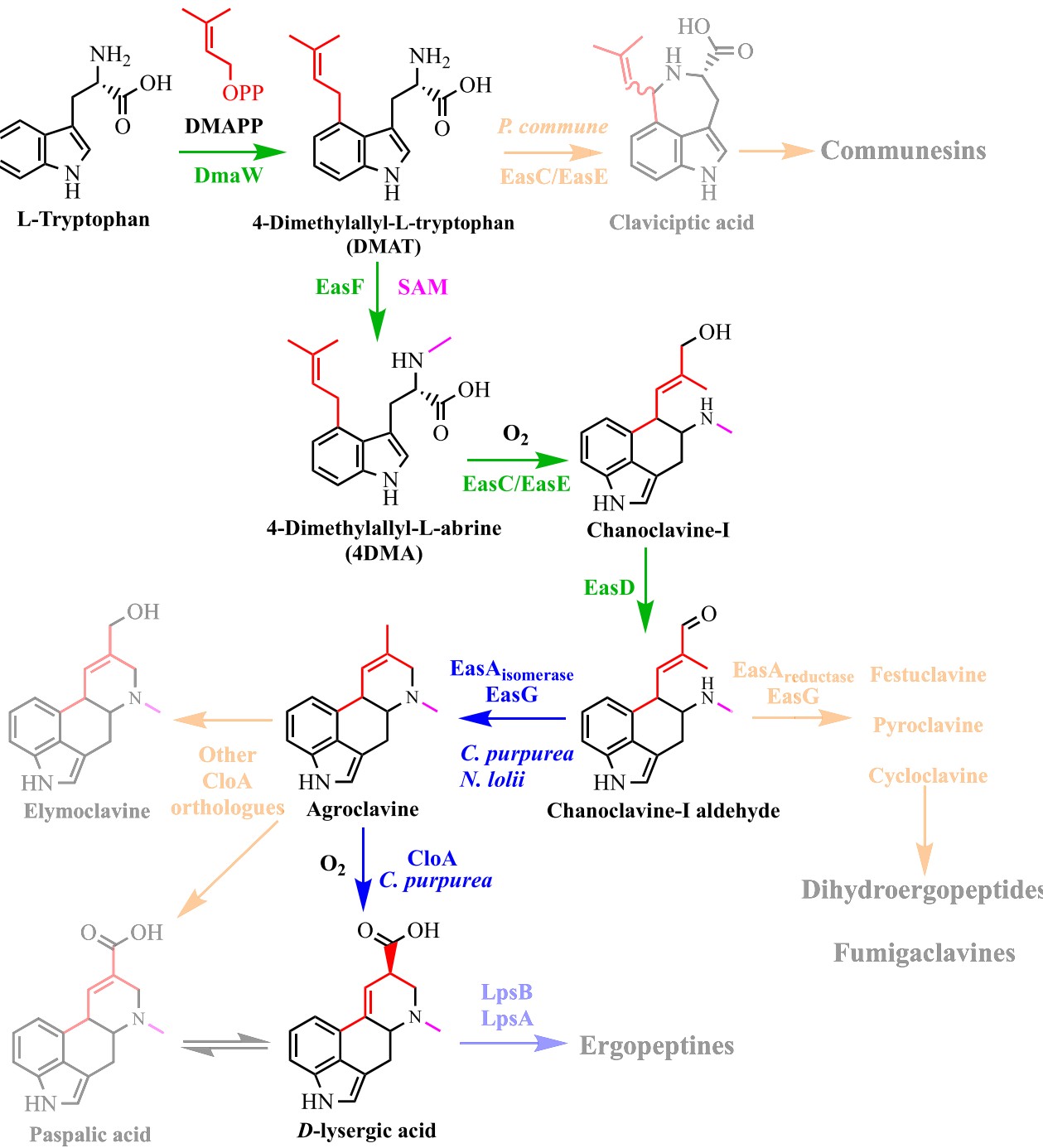

**Fig. 1 The complete biosynthetic pathway to the ergopeptides.** Beginning with tryptophan through the key intermediate and the focus of this work, DLA. Alternative branches of the ergot pathway are indicated by the faded regions. Colored arrows indicate the stages of the pathway. Green: early pathway common to all ergot producing species, leading to the first major branch point of the pathway at chanoclavine-I-aldehyde. The early pathway consists of four steps, requiring five enzymes (DmaW, EasF, EasC, EasE, and EasD). Blue: middle and late pathway accounting for the diversification of ergoline products found in the various lineages of ergot producing species. This stage of the pathway consists of two steps, requiring at least three enzymes (EasA, EasG, and CloA). Orange: Alternative branches of the pathway that lead to other ergoline derivatives.

Further comparison of the MS/MS fragmentation spectra confirmed the identity of the eluted compound to be agroclavine (Supplementary Fig. 8). The amounts of agroclavine produced by the strains expressing the different EasA orthologues showed EasA_Ec, EasA_Cp, and EasA_Nl, were all producing approximately 2.8–3.1 µg/L, while the strain containing the EasA_Pi orthologue was producing around seven times less agroclavine (Supplementary Fig. 9), and none of the selected EasA orthologs were found to be producing any of the alternative products. These

results identified three equally suitable EasA candidates that could be used to build the DLA pathway; for simplicity down the line we selected EasA_Ec for further pathway construction.

**Screening for a functional agroclavine oxidase to produce DLA.** To complete the DLA biosynthetic pathway, we addressed the oxidation of agroclavine to DLA. This two-step oxidation was proposed to be catalyzed by a cytochrome P450 (CYP450)

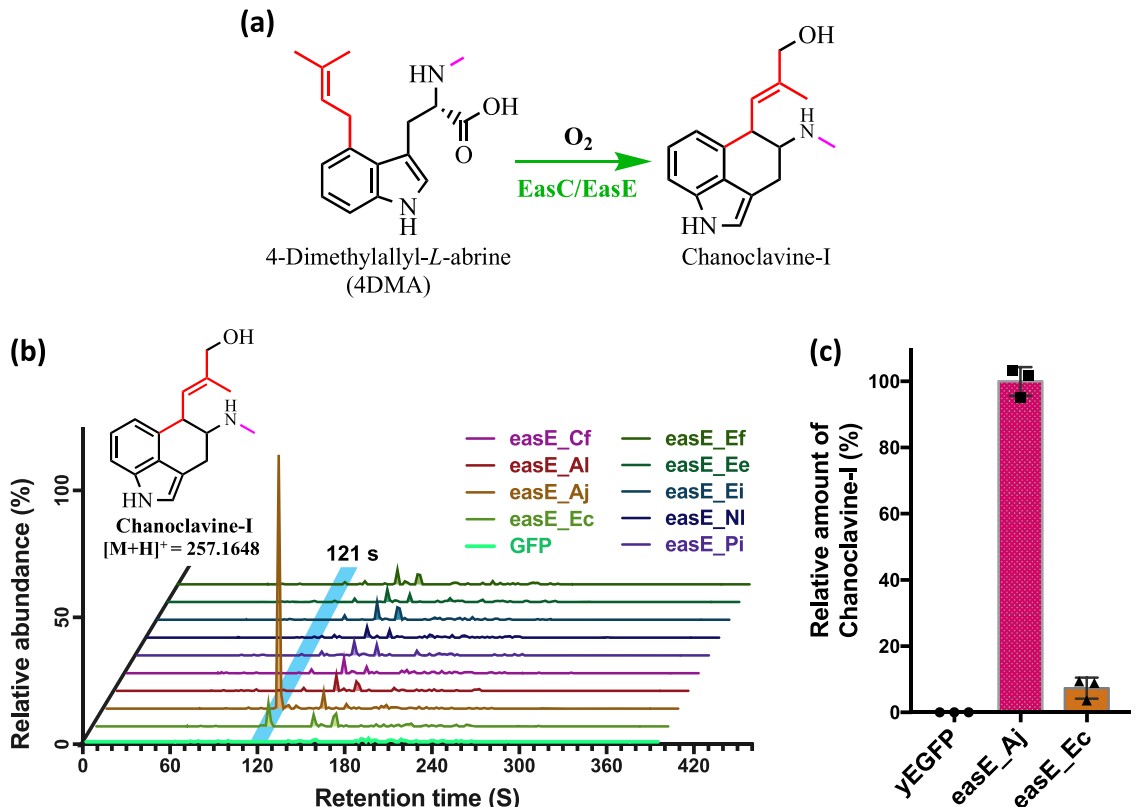

**Fig. 2 Screening of EasE orthologues with a yeast screening strain (YMC17) containing the genes *dmaW*, *easF*, and *easC* stably integrated onto the yeast genome.** Introduction of an episomal plasmid expressing the various EasE orthologues identified, to the strain allows for an easy screening platform. **a** Biosynthetic reaction producing chanoclavine-I from 4DMA. **b** LC-MS/MS extracted ion chromatograms of the screened EasE orthologues for the ion transition of 257 → 226 m/z, indicating the production of chanoclavine-I. Region highlighted in blue for the peaks eluting at 121 s, indicating the production of chanoclavine-I. Chromatograms are ordered (bottom to top): GFP control, easE_Ec, easE_Aj, easE_Al, easE_Cf, easE_Pi, easE_Nl, easE_Ei, easE_Ee, easE_Ef. **c** Relative amounts of chanoclavine-I produced by EasE_Aj and EasE_Ec estimated by the peak area from the MS/MS transition of 257 to 226 m/z. Data are presented as mean values +/− standard deviation. Error bars represent standard deviations calculated from three biological replicates. Source data are provided as a Source Data file.

monooxygenase, clavine oxidase (CloA)[27], though the mechanism of the reaction has yet to be biochemically characterized. From the myriad of ergot alkaloid producing fungi, a diverse set of oxidized agroclavine products have been isolated. These products correspond to products that have undergone either a single oxidation (elymoclavine/lysergol: $[M + H]^+ = 255$ m/z) or a double oxidation (paspalic acid/ DLA: $[M + H]^+ = 269$ m/z) with the possibility of an isomerization of the C8-C9 double bond (elymoclavine/ paspalic acid) to the C9-C10 position (lysergol/ DLA) (Fig. 4a).

We proceeded to identify an orthologue that expresses in yeast and specifically produces DLA. With an SSN generated from the CloA sequence predicted from *C. purpurea*, we were able to identify 15 orthologues from a cluster that consisted of sequences from organisms where DLA has been isolated from (Supplementary Discussion 1). These 15 orthologues were screened for functional expression in yeast (Fig. 4b). When provided with the agroclavine substrate, 5 of the 15 orthologues were found to catalyze the production of a molecule with a $[M + H]^+ = 269$ m/z that shared the same retention time as the DLA standard (Fig. 4c). The molecule also had identical MS/MS fragmentation spectra as a DLA standard, confirming that the molecule was DLA (Supplementary Fig. 10).

This screen has thus identified five orthologues of CloA (*C. pur.*, *C. pas.*, *N. lol.*, *E. coe.*, *P. ipo.*) that we could use for pathway construction. The top two producers (*C. pur.*, and *E. coe.*), and the worst producer (*P. ipo.*) from this screen (Fig. 4d, e) were then

further tested for DLA production in the context of an agroclavine-producing host (Supplementary Fig. 11). The *C. purpurea* and *E. coenophialia* orthologues were found to produce comparable levels of DLA in this context (Supplementary Fig. 11) and we selected the *C. purpurea* orthologue for incorporation into our strain due to lower variability in DLA titers.

**Assembling the components of the complete DLA biosynthetic pathway in yeast.** Equipped with a functional set of enzymatic and genetic elements for the reconstitution of the DLA biosynthetic pathway in yeast, we sought to construct a prototype DLA-biosynthetic strain. We decided to model our initial prototype after the strain that had been developed for the production of cycloclavine[28] that reported achieving an admirable yield of $529\ mg\ L^{-1}$. Our prototype design used stronger promoters for the less functional enzymes, such as *easE*, and multiple copies of the other pathway enzymes driven by weaker promoters in an attempt to attenuate the effects metabolic burden. We also included additional copies of the genes *fad1* and *pdi1* from yeast, which have been shown to aid protein folding and enhance the production of flavin adenine dinucleotide (FAD), a key co-factor for EasE activity[17].

Following the modified YeastFab pipeline (Supplementary Fig. 1, Supplementary Discussion 2), the prototype strain was sequentially constructed and expanded as four segments (Supplementary Fig. 12). Each segment was designed to produce intermediate products along the DLA biosynthetic route that can

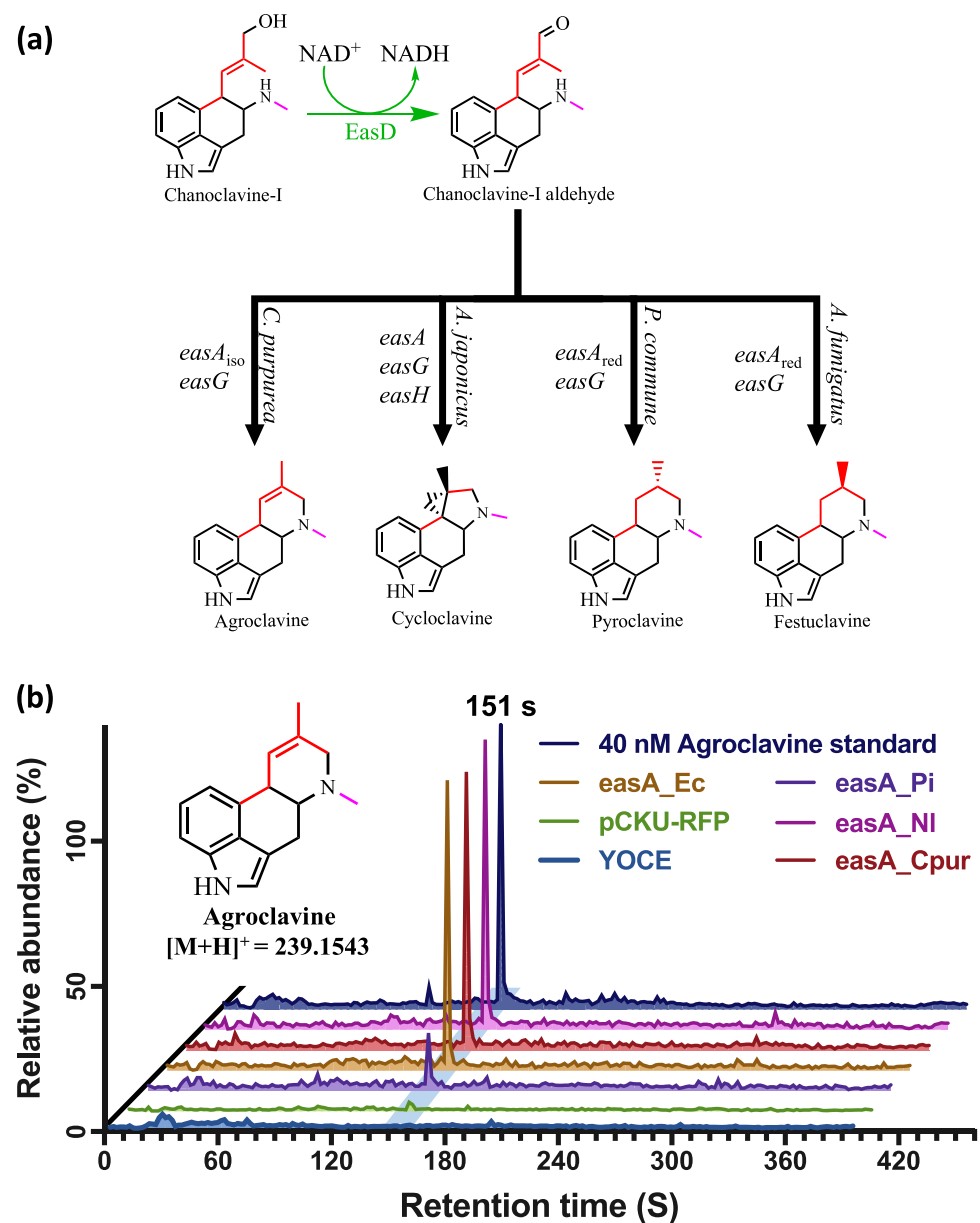

**Fig. 3 Screening of EasA orthologues. a** Reactions forming the ergoline D ring from chanoclavine-I. Different combinations of enzymes and isoforms (EasA, EasG, and EasH) control the divergence of the tetracyclic ergoclavine products formed at this branchpoint. Orthologues of EasA were screened with episomal plasmids in a strain containing genes for the early pathway enzymes *dmaW*, *easF*, *easC*, and *easE* integrated onto the yeast genome and regulated by the P$_{tef2}$, P$_{gpm1}$, P$_{gal10}$, and P$_{gal1}$ promoters respectively. **b** LC-MS chromatograms of the screened EasA orthologues. All selected orthologues were found to produce a compound that co-elutes with the commercially obtained agroclavine standard. Chromatograms appear in the order (bottom to top): YOCE, pCKU-RFP, easA_Pi, easA_Ec, easA_Cpur, easA_Nl, Agroclavine standard. Source data are provided as a Source Data file.

be easily detectable, namely chanoclavine-I, agroclavine, and DLA. This approach facilitated troubleshooting as well as enabling each intermediate strain to serve as a negative control for subsequent strains. Introduction of the first two segments (AgcM1B, AgcM2B; Supplementary Table 9) containing the genes for the early ergot pathway resulted in the production of chanoclavine-I (Fig. 5a). Increasing the copy numbers of *easC* and *dmaW*, as well as the supplemental expression of *fad1* and *pdi1*, improved chanoclavine-I yields as reported previously[17]. In the third segment, we introduced *easG*, *easA_Ec*, and further additional copies of *easF*, *easC*, and *easD*. This strain, AgcM33B, with three segments integrated into the yeast genome, produced a compound with a $[M + H]^+ = 239$ m/z, that co-elutes with the agroclavine standard (Fig. 5b) and confirmed to be agroclavine by

MS/MS fragmentation (Supplementary Fig. 13). The final reaction to DLA was then incorporated by the integration of *cloA_Cpur* as the last segment into the *ARS208* site. LC-MS analysis of the spent media from this strain (DLAM33B; Supplementary Table 9) showed the production of a compound with a $[M + H]^+ = 269$ m/z, that co-eluted with the DLA standard (Fig. 5c) and was further confirmed by MS/MS fragmentation (Supplementary Fig. 14).

Next, we validated the reconstitution of the pathway producing DLA by supplying our DLAM33B strain with $^{13}$C-2-indole-*L*-tryptophan ($^{13}$C-W) as a feedstock (Supplementary Fig. 15a), to track the progression of tryptophan through the reconstituted pathway. In these experiments, a shift of 1 m/z was detected in the chromatogram peak corresponding to DLA and in greater

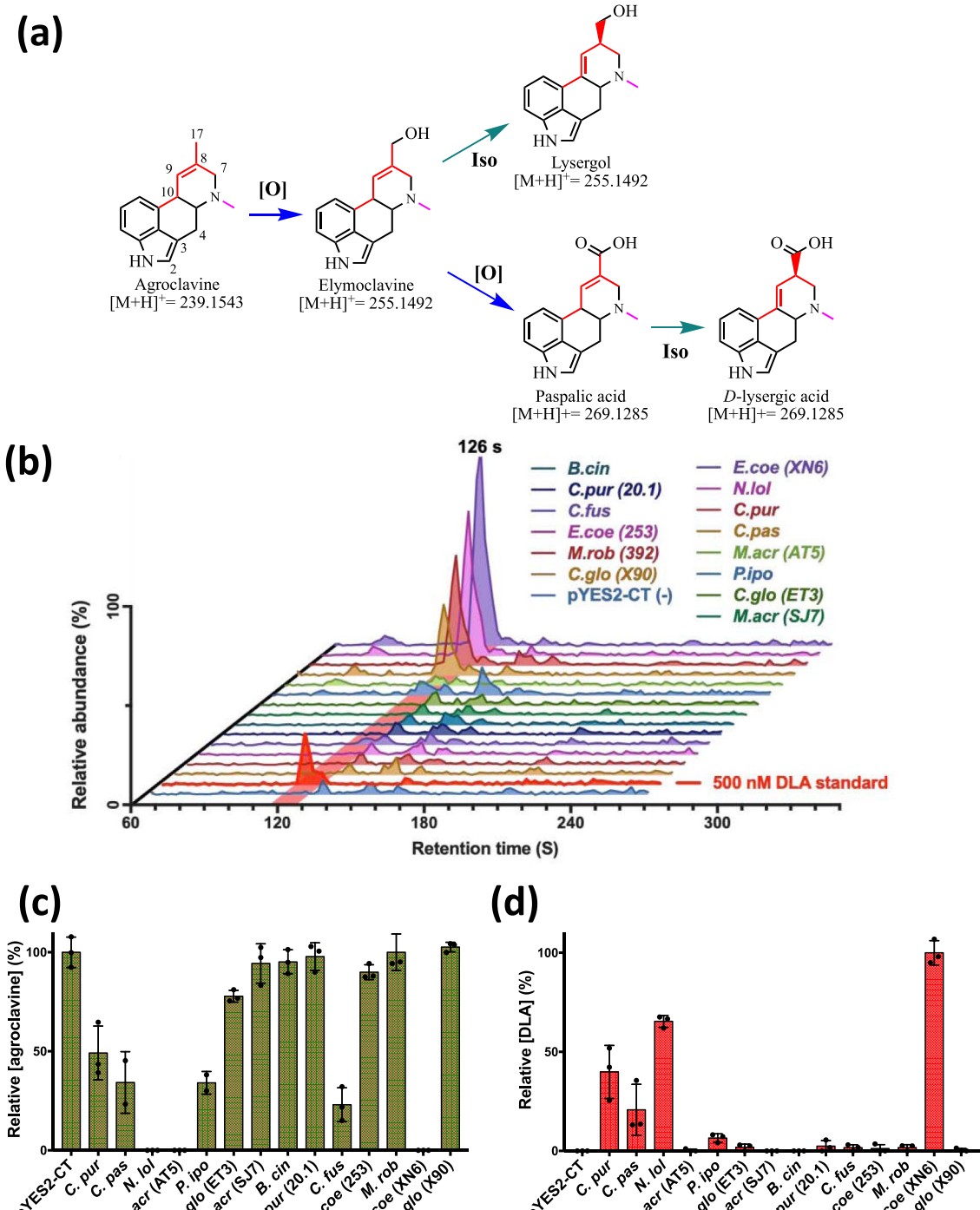

**Fig. 4 Screening of CloA orthologues. a** The putative oxidation and isomerization reactions catalyzed by CloA. **b** LC-MS/MS chromatograms of the products showing the ion transition of 269 → 223 m/z, produced by the CloA orthologs. Chromatograms appear in the order (bottom to top): pYES2-CT, DLA standard, *C.glo* (X90), *M.rob* (392), *E.coe* (253), *C.fus*, *C.pur* (20.1), B.cin, *M.acr* (SJ7), *C.glo* (ET3), *P.ipo*, *M.acr* (AT5), *C.pas*, *C.pur*, *N.lol*, *E.coe* (XN6). **c** Relative amounts of agroclavine consumed by the screened CloA orthologues. **d** Relative amounts of DLA produced by the various CloA orthologues. Data are presented as mean values + /− standard deviation. Error bars represent standard deviations calculated from three biological replicates. Source data are provided as a Source Data file.

abundance than expected for naturally occurring $^{13}$C-DLA (Supplementary Fig. 15b). The same mass shift was observed for all the intermediates of the DLA biosynthesis pathway (Supplementary Fig. 16). These data corroborated with the expected $[^{13}C\text{-}M + H]^+$ values incorporating $^{13}$C-W into the pathway and subsequent turnover into $^{13}$C-DLA and its $^{13}$C-intermediates. A comparison of the mass spectral patterns for

the proposed $^{13}$C-DLA further illustrated this +1/z shift across all dominant fragmentation peaks (Supplementary Fig. 16c) and indicated the utilization of tryptophan to produce DLA in the introduced pathway. Considering that the base strain of *S. cerevisiae* used in this work is not tryptophan auxotrophic, this experiment indicates that our engineered strain can produce DLA from tryptophan derived from central metabolism or the culture

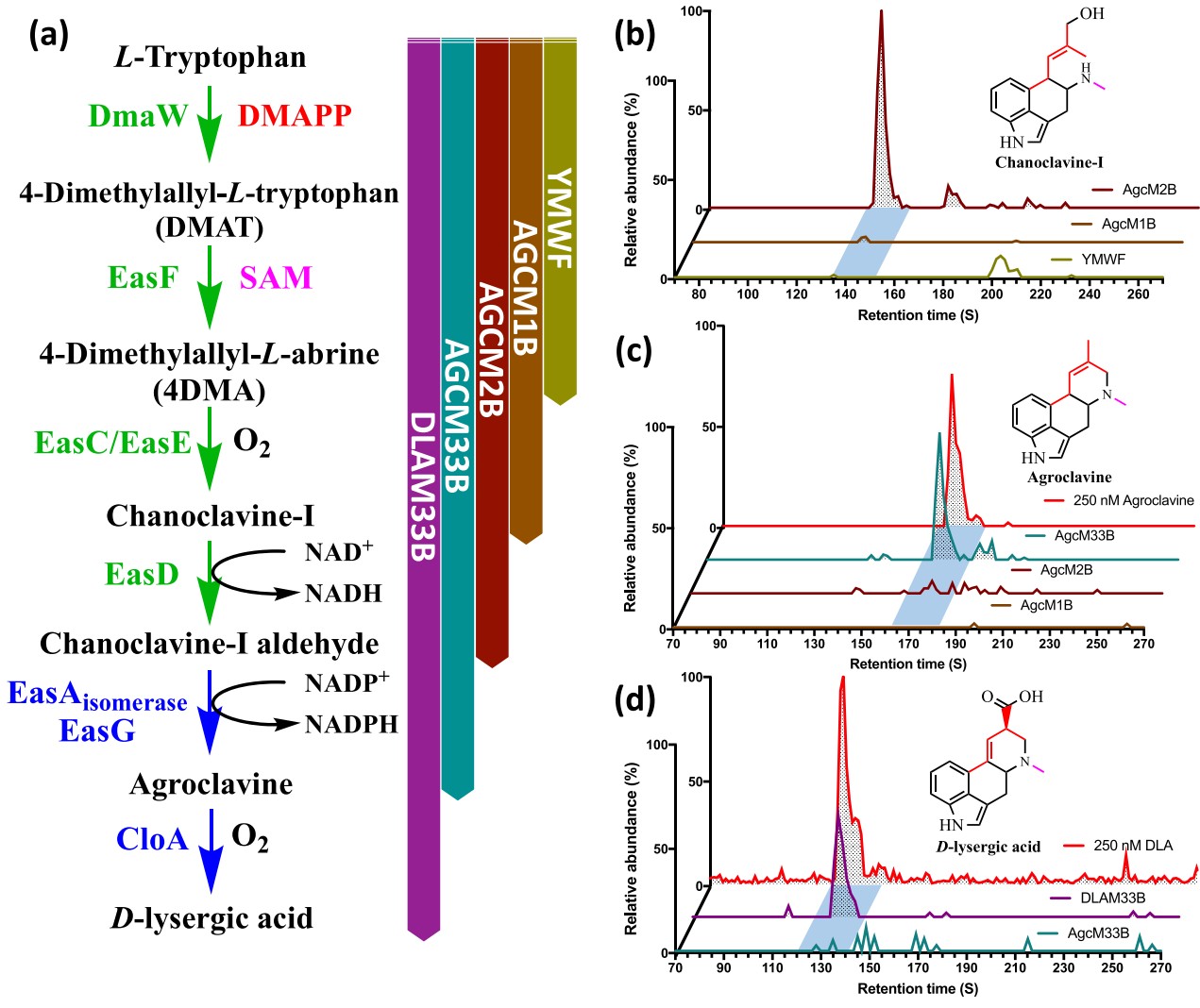

**Fig. 5 Assembling the functional parts into a DLA-producing yeast strain. a** Stepwise extension approach towards pathway construction in our engineered strains, enabling the use of each prior strain as a control for subsequent strains. **b** LC-MS/MS extracted ion chromatogram monitoring for the ion transition of $[M + H]^+ = 257 \rightarrow 226$ m/z, showing the production of chanoclavine-I in both AgcM1B and AgcM2B. **c** LC-MS/MS extracted ion chromatogram for the ion transition of $[M + H]^+ = 239 \rightarrow 208$ m/z, showing the production of agroclavine from AgcM33B. **d** LC-MS/MS extracted ion chromatogram showing the detected ion transition of $[M + H]^+ = 269 \rightarrow 223$ m/z, showing the production of DLA from DLAM33B. Peaks eluted at the time segments in blue correspond to the targeted compounds. Source data are provided as a Source Data file.

media. The next quantum to validate the production of DLA is by NMR analysis. To achieve this, we performed an 8 L scale-up of the shake-flask process to obtain the material required. The chemical shifts of the concentrated extracts were assigned based on predicted values and compared against the DLA standard (Supplementary Table 11).

The endpoint production titer of DLAM33B strain in small-scale shake-flask conditions measured to be 71.5 $\mu$g L$^{-1}$ (Supplementary Fig. 19; Supplementary Table 8). To demonstrate the scalable application of our engineered strain, we further attempted 1 and 4 L scale bioreactor fermentations. In these fermentation experiments with the improvement of culture oxygenation, carbon source, and inducer control (galactose was used as both carbon source and expression induction agent), as well as pH control, we were able to achieve sustained cell proliferation to a maximum cell density of around 29 g L$^{-1}$ and improved production titers (Fig. 6), attaining 1.7 mg L$^{-1}$ and 1.4 mg L$^{-1}$ at the endpoints for the 1 and 4 L fermentations, respectively (Supplementary Table 12). With the existing

parameters, performing a similar fed-batch cultivation at the 1000 L scale would produce 1.5 g per reactor in approximately a week and a half, and in a year, this would produce 52 g of DLA (with extrapolation, a 100-tonne bioreactor facility would produce 5.2 tonnes of DLA annually); such a level of production is still a preamble to something scalable to the 10-15 tonnes required annually. It is however conceivable that with further bioprocessing and strain optimization, commercial level titers could be attainable.

This work builds on the growing body of work demonstrating the use of industrially tractable microorganisms for the production of complex natural products using economical and renewable feedstocks, such as what has been done for the biosynthesis of the opioids[29]. Engineered strains provide an excellent platform to drive the discovery of semi-synthetic therapeutic lead compounds, as well as for developing pilot strains for producing important naturally derived therapeutics.

Lastly, with the recent renaissance of research into repurposing psychedelic compounds for anti-depressives and anti-anxiolytics

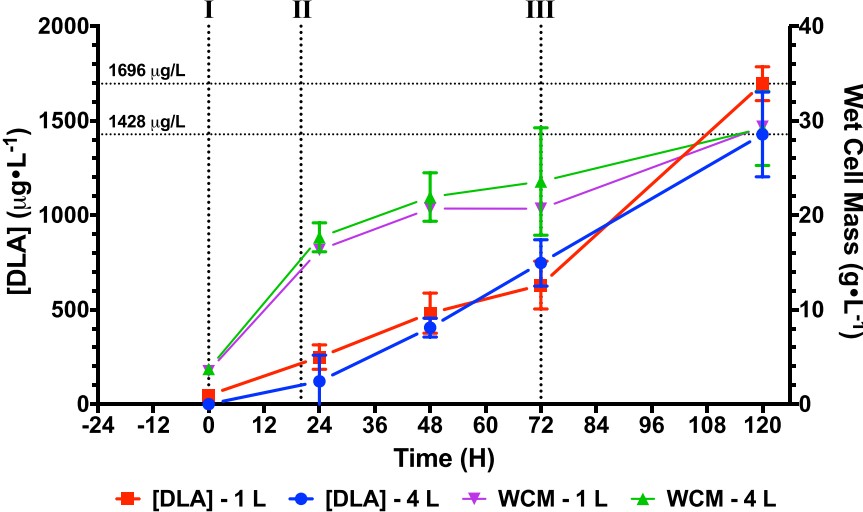

**Fig. 6 Production of DLA from DLAM33B in 1 and 4 L scale fermentation.** The fermentation process was modeled after the induction protocol used in shake flask experiments, with additional galactose and 10X SC-URA media supplemented at defined feeding phases (I, II, III) in a fed-batch mode. 50 mM Ammonium-succinate was supplemented into the culture media from the onset to maintain a pH of 5.8. Both 1 and 4 L culture fermentations achieved a maximum wet cell mass of around 26 g L⁻¹ and a maximum DLA titer of 1.7 mg L⁻¹ and 1.4 mg L⁻¹. Feeding phase I: initial induction phase mimicking the addition of galactose for induction in shake flask experiments. 10X feed solutions were added to the vessel to a final concentration of 1X, at a rate of 3.5 ml min⁻¹. Feeding phase II: sustained feeding phase, a second round of 10X feed solutions were supplemented to the culture to a final concentration of 1X over a period of 52 h. Feeding phase III: starvation phase, no additional carbon or nitrogen source was supplemented. Data are presented as mean values + / − standard deviation. Error bars were calculated from three biological replicates. Source data are provided as a Source Data file.

applications[30], we believe that our strain could be used to support efforts to probe the natural and semi-synthetic chemical space of ergoline-based therapeutics, to identify leads with enhanced therapeutic potential and fewer adverse effects.

In this work, we described the identification of alternative orthologues of the fastidious enzymes along the ergot alkaloid pathway, and along with it the development of customized strains for their systematic screening. Through this approach, central to the tenets of synthetic biology, we successfully identified candidates for the reconstitution of the pathway to DLA in *S. cerevisiae*. These were subsequently used to create an engineered yeast strain capable of producing DLA from central metabolism. While the titers achieved are not directly translatable into a commercial product, our work serves as a proof-of-concept for the complete heterologous biosynthesis of DLA from sugar. An engineered strain to produce the key ergoline-derivative API, DLA, eliminates the need to hydrolyze ergopeptines and minimizes purification and downstream processing complexity. Further strain optimization, aimed towards identifying and relieving bottlenecks in the pathway, as well as improved bioprocess optimization, would indubitably push DLA production titers towards the required levels for resolving the current challenges in its production.

## Methods

**Cultivation, media, and strains**. *E. coli* XL1Blue (Stratagene), *E. coli* NEB Stable (New England Biolabs), and *S. cerevisiae* strain BY4741 were the base strains used in this study. *E. coli* constructs were grown in lysogeny broth (LB) at 37 °C with the appropriate antibiotics. Competent *E. coli* cells were prepared and transformed following the Inoue protocol, with the modification that cells were grown at 30 °C to mid-logarithmic phase before further processing[31]. Yeast strains were grown in either Yeast extract-Peptone-Dextrose (YPD) or in Synthetic-Complete (SC) media omitting the appropriate nutrient for selection[32]. Transformation of plasmids and DNA fragments for chromosomal integration in *S. cerevisiae* were performed using the Lithium acetate/ PEG-3350/ single-stranded carrier DNA protocol[33].

**Golden-gate assembly of parts and pathways**. The Golden-gate assembly used in this study predominantly is in accordance with existing published protocols, save for a few modifications[34,35]. Reactions were prepared as 10 μL pots, each containing 1 μL 10X T4 ligase buffer (New England Biolabs), 1 μL 100X bovine serum albumin (New England Biolabs), 5 U of restriction enzyme (BsaI-HFv2 or Esp3I, New England Biolabs), 10 U T4 DNA ligase (New England Biolabs), 15 ng of destination plasmid, 1 μL per insert and brought to 10 μL with sterile deionized distilled water (ddH₂O). The assembly reactions were cycled beginning with 37 °C for 5 min followed by 25 °C for 10 min for 25 cycles, before finishing with 55 °C for 20 min and 80 °C for a further 20 min. The reaction mixture was subsequently directly used for the transformation of chemically competent XL1Blue (Level 0/1 constructs) or NEB Stable (Level 2 and beyond).

Yeast promoter and terminator parts were PCR amplified from *S. cerevisiae* S288C genomic DNA and cloned into HcKan_P and HcKan_T respectively. For simplicity, promoter and terminator sequences were defined as the region 500 base pairs (bp) upstream or downstream of the ORF used for naming the genetic element. The genes encoding the enzymes of the pathway were codon-optimized for yeast expression, synthesized, and cloned into HcKan_O. The genes for FAD1 and PDI1 were PCR amplified (Takara PrimeSTAR™) from *S. cerevisiae* S288C genomic DNA and cloned into HcKan_O. All level 0 and level 1 constructs assembled were verified by colony PCR using Taq DNA polymerase (New England Biolabs) and Sanger sequencing, while level 2 constructs and genome integration constructs were verified by colony PCR.

**Small-scale yeast culture for the production of ergot alkaloids**. For each assayed construct, three isolated colonies either freshly transformed or streaked were used to inoculate a 10 mL pre-culture in a 50 mL tube of the appropriate growth media and grown for at least 18 h at 30 °C in a shaking incubator at 210 rpm. These cultures were then back-diluted to a final OD₆₀₀ of 0.0125 in 10 mL of the appropriate SC media supplemented with 0.1% (w/v) *D*-glucose. Cultures were further grown to an OD₆₀₀ of 0.8 before galactose was added to a final concentration of 2% (w/v) from a filter-sterilized 20% (w/v) stock solution. The caps of the culture tubes were then replaced with autoclaved aluminum foil caps and grown for 120 h at 24 °C. Cultures were subsequently pelleted by centrifugation at 2236 × g for 10 min. A 1 mL aliquot of the collected supernatant from each sample was then filtered through a 0.2 um PTFE syringe filter and analyzed using liquid chromatography—tandem mass spectrometry (LC-MS/MS).

**Fed-batch fermentative production of DLA from engineered yeast**. Culture media used in both 4 and 1 L fermentations consisted primarily of SC-URA with 0.1% (w/v) glucose and 50 mM ammonium-succinate, pH 5.8 (SCUS). Two feed solutions were used, Feed 1 consisted of 10X SC-URA amino acids mix and 10X Yeast nitrogen bases; Feed 2 consisted of 20% (w/v) galactose and 100 mM ammonium-succinate, pH 5.8.

The seed culture was prepared by growing a single colony of DLAM33B from a freshly streaked plate in 10 mL SC-URA media with 2% (w/v) glucose at 30 °C overnight. Fermentation was initiated by inoculating the fermentation vessel (INFORS HT Minifors 2, Bottmingen, Basel, Switzerland) filled with 2 L or 500 mL of SCUS with the seed culture to a final OD₆₀₀ of 0.0125.

The fermentation process begins with an initial outgrowth phase (phase 0) at 30 °C with stirring at 1000 rpm and compressed air (Ekom DK40 2 V, Singapore) was used to supply an airflow of 1 vessel volume per minute (either 4 L min$^{-1}$ or 1 L min$^{-1}$), Dissolved oxygen (DO) level was maintained at >90 % saturation through an automated cascade to increase stirring rate up to 1500 rpm and to increase airflow up to 2 vessel volume per minute (8 L min$^{-1}$ or 2 L min$^{-1}$). This phase allows for the depletion of the glucose present and for the culture to propagate to an adequate cell density for induction. After 24 h, the induction phase (phase I) is initiated by pumping both Feed 1 and 2 into the vessel at 3.5 mL min$^{-1}$, until the volume of each feed pumped in to the starting fermentation culture volume is 1 part to 8 parts. The temperature was also lowered to 24 °C for the induction of the pathway. Phase 0 and I directly mimic the conditions used for pathway induction at the shake flask scale. Phase II was pre-programmed to begin after 20 h of phase I, when the galactose supplemented in phase I is expected to begin it's exponential decline[36]. In this phase, a steady low-level flow rate of 0.05 mL min$^{-1}$ (1 L) or 0.18 mL min$^{-1}$ (4 L) for both Feed 1 and 2 is maintained over 28 h to maintain sufficient nutrients in the culture, as well as ensure sufficient expression of pathway genes. At phase III or the starvation phase, no additional feed was supplemented, and the fermentation was allowed to carry on for an additional two days.

The fermentation was monitored by drawing 10 mL samples daily and assessed for wet cell mass (WCM) and DLA production titer. DLA production titer was measured by standard addition. Briefly, samples were spiked with 0, 0.2, 1, 2, 5, 8, 10, and 15 uM of DLA standard and analyzed on the LC-MS. A linear regression was then performed for each sample analyzed and the concentration of DLA required to double the area of the DLA peak was used to determine the concentration of DLA in the analyzed samples.

**Screening of *cloA* orthologues.** Genes encoding the various *cloA* orthologues selected were synthesized and cloned (BioBasic) into the pYES2/CT vector (Invitrogen). Cells were cultured and expression was induced as described in small-scale yeast culture for the production of ergot alkaloids, with the exception that a 1 mL aliquot of induced cells were removed from each tube and transferred into a 1.5 mL microcentrifuge tube for use in this assay. Cells were then pelleted by centrifugation at 21,000 *g* for 1 min. The pellet was then resuspended in 1 mL of phosphate buffered saline (PBS), pH 7.4. Agroclavine was then spiked into each tube to a final concentration of 5 *µ*M and incubated at 30 °C overnight. The PBS incubations were then pelleted by centrifugation and the supernatant from each sample was filter sterilized using a 0.2 um PTFE syringe filter and similarly analyzed using liquid chromatography–tandem mass spectrometry (LC-MS/MS).

**Analysis of ergot alkaloids by high-performance liquid chromatography–tandem mass spectrometry (HPLC-MS/MS).** All samples were analyzed using the Agilent 1290 Infinity LC system coupled to an Agilent 6550 iFunnel QTOF with an electrospray ionization source. Samples were separated using an Agilent InfinityLab Poroshell 120 EC-C18 column with the dimensions of 2.1 × 100 mm, 1.9 *µ*m particle size. The mobile phases used consisted of: A, containing water with 0.1% formic acid; and B, containing acetonitrile with 0.1% formic acid. Chromatography was carried out over a constant flow rate of 0.5 mL/min, 1 *µ*L injection volume, with a stepped gradient as follows: 95% A/5% B for 0.6 min, 65% A/35% B to 2.6 min, 1% A/99% B to 4.6 min. The column was washed with 100% B for 2 min before re-equilibrating to 95% A/5% B for 1 min.

Mass data acquisition was set to targeted MS/MS mode using fixed polarity (positive), from the eluent beginning from 1 min into the run and ending at 4.5 min. Instrument parameters were set to run at; source gas temperature and flow of 200 °C and 10 L/min, sheath gas temperature and flow of 350 °C and 10 L/min, nebulizer pressure at 50 psig. Capillary and nozzle voltages were set to 4000 V and 0 V respectively. MS1 was set to a mass range of 40–1000 m/z, at a scan rate of 3 spectra/second. MS2 was set to a mass range of 40–1000 m/z, at a scan rate of 6 spectra/second using a fixed collision energy of either 10 *e*V for screening experiments or 20 *e*V for the analysis of the reconstituted strains.

The targeted mass for the screening of easE orthologs to produce chanoclavine-I was set to 257.1648 m/z, on a narrow isolation bandwidth (1.3 amu). In the screen for isomerase variants of easA, the targeted mass was set to 239.1543 m/z, on a narrow isolation width (1.3 amu). For the screening of *cloA* orthologs, the targeted masses were set for: 1) 239.1543 m/z, narrow isolation width (1.3 amu); 2) 269.1285 m/z, narrow isolation width (1.3 amu). In the assay for the reconstitution of the ergot alkaloid pathway, the targeted masses were set for: 1) 287.1754 m/z, narrow isolation width (1.3 amu); 2) 257.1648 m/z, narrow isolation width (1.3 amu); 3) 239.1543 m/z, narrow isolation width; 4) 269.1285 m/z, narrow isolation width (1.3 amu). To detect the incorporation of $^{13}$C-labelled-tryptophan into the reconstituted pathway, the targeted masses were set for: 1) 269.1285 m/z, narrow isolation width (1.3 amu), 2) 270.1318 m/z, narrow isolation width. All data analysis and instrument control were performed using the Mass Hunter software suite (Agilent).

Determination of compound identities were performed by the comparison of retention time and MS/MS product ion spectrum against commercially obtainable standards where available (*D*-lysergic acid; Chiron) (Agroclavine; Chiron/Toronto Research Chemicals). Determination of 4DMAT, 4DMA and chanoclavine-I was performed by the comparison of the retention times and MS/MS product ion spectrum against in vitro biosynthesized products of purified *dmaW*, *easF*, *easC* and cell extracts expressing easE_Aj. Quantification of agroclavine and DLA

produced was performed using either a calibration curve established from the commercial standards or by standard addition. Agroclavine was quantified by monitoring the transition of the 239 m/z precursor ion to 208 m/z and DLA was quantified by monitoring the transition of the 269 m/z precursor ion to 223 m/z. Linear regression analysis of the standard curves were performed using Graphpad Prism version 7.00 for Windows (Graphpad Software, San Diego, CA).

**Reporting Summary**. Further information on research design is available in the Nature Research Reporting Summary linked to this article.

## Data availability

All data is available in the main text or the supplementary materials. Source data are provided with this paper.

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

## Acknowledgements

The authors wish to acknowledge the assistance of Dr. Wei Zhe Teo at the BioFoundry Singapore (hosted by the NUS Synthetic Biology for Clinical and Technological Innovation) and the staff at the London BioFoundry. We also thank Ms. Yanhui Han and Mr. Gustavo Giraldi Shimamoto for the NMR analytical service at the Department of Chemistry, National University of Singapore. This work was supported by the National Research Foundation Singapore (R-183-000-418-592 to WSY) and by UK Research and Innovation (EPSRC grants EP/L011573/1; EP/K038648/1 to PF).

## Author contributions

P.F. and W.S.Y. conceptualized the manuscript. G.W. and Y.Q.T. performed the expansion of the synthetic biology toolkit. G.W. performed the enzymology work for the pathway up to agroclavine production. G.W. and L.R.L. performed the enzymology work for the transformation of agroclavine to DLA. G.W., D.B., and M.K.G. analyzed the mass spectrometry data. G.W., M.K.G., P.F., and W.S.Y. wrote the manuscript.

## Competing interests

The authors declare no competing interests.
