## [Peer Review File · Nature Communications]

Reviewers' Comments:

Reviewer #1:

Remarks to the Author:

This manuscript report the reconstituting the complete biosynthesis of the D-lysergic acid in yeast by gene identification and pathway engineering, which enabled production of 71.5 µg/L D-lysergic acid in small-scale shake flasks. This study shows a good example to heterologous reconstruction of ergot alkaloid biosynthesis pathway in yeast. There are some concerns should be addressed for publication

1. This study claims (several time in the manuscript) the use of bioinformatic tools and systematic synthetic biology methods can be used to establish a platform towards the scalable production DLA and related ergot alkaloids from microbial fermentations. Please soft the claim. In this manuscript, there is neither new bioinformatic tools and systematic synthetic biology methods described in this manuscript (just used methods described previously), nor high-level production of DLA that make no sense for scalable production.
2. The authors constructed DLAM33B for D-lysergic acid biosynthesis with integration of the whole biosynthetic pathway. Only the D-lysergic acid titer and spectrum was showed in Fig. 5C. Were there some intermediates detected. This data would be helpful for pathway optimization
3. In the abstract, the authors claimed that the reconstituting the complete biosynthetic pathway enabled de novo production of 71.5 µg/L D-lysergic from glucose. I however found galactose was added to a final concentration of 2% (w/v) for yeast cultivation. Please check this.
4. There is little information of P450 gene cloA. Is this gene first identified in this study? Which kind of P450 it is? Need some cofactor for activity in yeast? Please give some more analysis and information of CloA, since this enzyme is very important for DLA biosynthesis.
5. In Fig. 3, the authors screened several version of easA orthologues for agroclavine biosynthesis, and found easA_Ec, easA_Cp, or easA_NI, all successfully produced similar amount of agroclavine. Which is interesting since different version of easA contributed to synthesis of different compounds as show in Fig. 3A. Which enzyme control the branch synthesis of different compounds, easA, easG or easH? Please discuss more about this. Is there some by-products other than agroclavine when using different version of easA?

Reviewer #2:

Remarks to the Author:

The authors described the identification of several enzymes for D-lysergic acid biosynthesis from fungi. They engineered yeast strains and introduced the identified enzymes lead to a titer of 71.5 µg/L in shake flask. The work in interesting, but the description of the results in this manuscript can be improved.

In the abstract and the introduction part, the authors can describe more about the development of metabolic engineering and synthetic biology strategies used for D-lysergic acid and other alkaloids. These two parts can be improved.

The identification of the enzymes is essential for DLA biosynthesis in engineered yeasts, but the authors described little about enzyme recovery in the manuscript. Even in the methods part, the authors only gave limited information. The authors can strength this part in the manuscript.

The authors can link the supplementary information and the main body of this manuscript, which might help the readers to understand this manuscript.

In Fig. 1, the author described the complete biosynthetic pathway of the ergot alkaloids. However, the authors described little about the derivatives. Fig.1 needs to be focus on the DLA biosynthetic pathway.

Line 104 where does this easE_Aj come from? No description of this gene

Line 115-116, easE_Aj was not described in previous content.

Fig 2C, the colors of ease_Ec and easE_AI are hard to read.

The cartoon description of Fig. 2A, Fig.3B, Fig. 4B, Fig. 5A, 5B, 5C, Fig.6A provided little information, they should be moved to the supplementary files. Giving strain details is enough.

In the conclusion part, most description in the 1st paragraph (line 236-245) is not related to the

biosynthesis of DLA. This should not be in the conclusion part. The same to paragraph 2 of the conclusion part, the authors need to improve this conclusions part.

We have noted the reviewers' suggestions for improvement and put forth our responses below. Comments from separate reviewers are listed below and our replies are in blue.

Editorial team's Comments:

Your revision should address all the points raised by our reviewers (see their reports below). **In particular, the editorial team would be interested in seeing the concerns of Reviewer #1 addressed through demonstration of productivity in larger volume fermentators.**

We thank the editorial team for this helpful suggestion and refer them to our response to the first point of Reviewer #1, where we have included new data for 1 and 4 L fermentations in this manuscript.

REVIEWER COMMENTS

Reviewer #1 (Remarks to the Author):

This manuscript report the reconstituting the complete biosynthesis of the D-lysergic acid in yeast by gene identification and pathway engineering, which enabled production of 71.5 µg/L D-lysergic acid in small-scale shake flasks. This study shows a good example to heterologous reconstruction of ergot alkaloid biosynthesis pathway in yeast. There are some concerns should be addressed for publication

1. This study claims (several time in the manuscript) the use of bioinformatic tools and systematic synthetic biology methods can be used to establish a platform towards the scalable production DLA and related ergot alkaloids from microbial fermentations. **Please soft the claim.** In this manuscript, there is neither new bioinformatic tools and systematic synthetic biology methods described in this manuscript (just used methods described previously), nor high-level production of DLA that make no sense for scalable production.

We appreciate the reviewer's comment and have re-written the manuscript to soften these claims. As per the recommendation of the editor as well, have also included data from larger scale fermentations (1 and 4 L) to better substantiate our claim for scalable production (Fig. 6).

2. The authors constructed DLAM33B for D-lysergic acid biosynthesis with integration of the whole biosynthetic pathway. Only the D-lysergic acid titer and spectrum was showed in Fig. 5C. **Were there some intermediates detected.** This data would be helpful for pathway optimization

We thank the reviewer for the helpful suggestion and agree that such data would be essential for performing metabolic flux analysis that is crucial for pathway optimization. We have detected the pathway intermediates and would like to direct the reviewer to supplementary figures Fig. S15 and Fig. S16, where we have traced the progression of ¹³C-labelled tryptophan through the reconstituted pathway. All intermediates along the pathway to DLA from tryptophan could be detected, both with ¹³C labelling and without (data not shown). Apart from agroclavine however, all other intermediates could not be quantified due to the lack of available commercial standards. Further work on pathway optimization would see the purification of the intermediates for use as calibration standards, but as the focus of this report is the creation of a proof-of-concept strain, it was not included.

3. In the abstract, the authors claimed that the reconstituting the complete biosynthetic pathway enabled de novo production of 71.5 µg/L D-lysergic from glucose. **I however found galactose was added to a final concentration of 2% (w/v) for yeast cultivation.** Please check this.

We thank the reviewer for pointing out the oversight on our part. We have since amended the manuscript to either "...from central metabolism." Or "...from simple sugars." instead of "...from glucose".

4. There is little information of P450 gene CloA. Is this gene first identified in this study? Which kind of P450 it is? Need some cofactor for activity in yeast? **Please give some more analysis and information of CloA**, since this enzyme is very important for DLA biosynthesis.

We thank the reviewer for the insightful comment.

This study was not the first study to identify the P450 gene, CloA. It was first proposed that a mixed function oxygenase, was involved in the oxidation reaction of agroclavine to elymoclavine, after microsomal preparations of the strain *C. purpurea* PRL 1980, was found to be oxidise agroclavine to elymoclavine in an oxygen and NADPH dependent reaction (Hsu and Anderson, 1970). Separately, it was observed that the same oxidation reaction of agroclavine by microsomal fractions of *C. purpurea* PRL 1980, was inhibited by carbon and that the photochemical action spectrum showed an absorbance maximum at 450 nm, which is characteristic of cytochrome P450 enzymes (Kim et al., 1981).

The gene encoding a putative cytochrome P450 in the ergot alkaloid synthesis (EAS) gene cluster of *C. purpurea* strain P1, was first identified and designated clavine oxidase A (CloA) by Haarmann et al., (Haarmann et al., 2005). Disruption of the gene encoding CloA in *C. purpurea* strain P1, resulted in a strain that accumulated agroclavine and elymoclavine (Haarmann et al., 2006). The authors proposed that CloA was an elymoclavine monooxygenase, and that a separate enzyme catalyses the oxidation of agroclavine to elymoclavine. (Haarmann et al., 2006).

The oxidation of agroclavine to lysergic acid was also observed when CloA from *Epichloe* species Lp1, was heterologously expressed in *A. fumigatus* (Robinson and Panaccione, 2014). This led the authors to propose that CloA, catalyses a two-step oxidation reaction of agroclavine to paspalic acid, followed by an isomerization of the double bond of the ergoline ring from the Δ C8-C9 to the Δ C9-C10 position (Robinson and Panaccione, 2014).

The lack of information regarding CloA and the contradicting observations made by Haarmann et al. and Robinson and Panaccione, led us to conduct an investigation in to the activity of CloA.

A separate manuscript is in the process of preparation, detailing the activity of the cytochrome P450 enzyme CloA in *S. cerevisiae* and examining regions of the enzyme that are critical to its enzymatic function. Briefly, we found that CloA activity was improved by supplementation with iron (III) chloride and 5-aminolevulinic acid (heme precursor), which suggests that a heme prosthetic group is present. We will also discuss in the manuscript, that the one-step oxidation of agroclavine to elymoclavine or its structural isomer lysergol, or a two-step oxidation reaction of agroclavine to paspalic acid or its structural isomer DLA may be controlled by several critical residues on the surface and within the substrate access channel of CloA.

1. Hsu, J. C., & Anderson, J. A. (1970). Agroclavine hydroxylase of *Claviceps purpurea*. *Journal of the Chemical Society D: Chemical Communications*, (20), 1318.

2. Kim, I.-S., Kim, S.-U., & Anderson, J. A. (1981). Microsomal agroclavine hydroxylase of *Claviceps* species. *Phytochemistry*, 20(10), 2311–2314.
3. Haarmann, T., Machado, C., Lübbe, Y., Correia, T., Schardl, C. L., Panaccione, D. G., & Tudzynski, P. (2005). The ergot alkaloid gene cluster in *Claviceps purpurea*: Extension of the cluster sequence and intra species evolution. *Phytochemistry*, 66(11), 1312–1320.
4. Haarmann, T., Ortel, I., Tudzynski, P., & Keller, U. (2006). Identification of the Cytochrome P450 Monooxygenase that Bridges the Clavine and Ergoline Alkaloid Pathways. *ChemBioChem*, 7(4), 645–652.
5. Robinson, S. L., & Panaccione, D. G. (2014). Heterologous Expression of Lysergic Acid and Novel Ergot Alkaloids in *Aspergillus fumigatus*. *Applied and Environmental Microbiology*, 80(20), 6465–6472.

5. In Fig. 3, the authors screened several version of easA orthologues for agroclavine biosynthesis, and found easA_Ec, easA_Cp, or easA_Nl, all successfully produced similar amount of agroclavine. Which is interesting since different version of easA contributed to synthesis of different compounds as show in Fig. 3A. **Which enzyme control the branch synthesis of different compounds, easA, easG or easH?** Please discuss more about this. **Is there some by-products other than agroclavine when using different version of easA?**

We thank the reviewer for the helpful suggestion and have revised the section “Identifying isomerase variants of easA” to explain the control of the structural branch point in more detail, as well as to indicate that no by-products were detected with any of the characterized easA orthologues.

In a more detailed description, easA is responsible for controlling the resulting product. An isomerase variant of easA will retain the double bond on the C8-C9 position of the ergoline ring after the re-orientation of the aldehyde group, thus producing agroclavine after easG reduces the iminium intermediate that was formed between the re-oriented aldehyde group and the methylamino group (Cheng, et al. (2010a); Cheng, et al. (2010b); Floss, et al., 1968; Matuschek, et al. (2011)). A reductase variant of easA on the other hand, will completely reduce the C8-C9 double bond to allow for the positioning of the aldehyde group in proximity of the methylamino group, thus producing pyroclavine or festuclavine depending on the stereochemistry of the methyl group on the C7 of the ergoline ring (Cheng, et al. (2010a); Cheng, et al. (2010b); Floss, et al., 1968). The presence of the easH orthologue from *Aspergillus japonicus* can intercept the iminium intermediate from a reductase variant of easA to re-arrange the 6-membered ergoline D ring into a cyclopropyl moiety that once reduced by easG, produces cycloclavine (Jakubczyk, et. al., (2015); Yan, L. and Y. Liu (2019)).

In the work by Cheng, J.Z., et. al., (2010), they reported the identification of a key residue at the structurally corresponding 176-position of easA that was responsible for determining if an isomer of easA had isomerase or reductase activities (Cheng, et al. (2010b)). They demonstrated the transformation of an isomerase variant into a reductase variant with a single point mutation of the F176 residue into a Y176 residue (Cheng, et al. (2010b)). We had therefore further checked the sequence alignments of the easA orthologues in our identified isomerase isofunctional cluster, and only selected sequences that contained a phenylalanine at the 176-position.

We have therefore opted to not expound into too much detail on the mechanistic background behind this branchpoint in the pathway, as these details will be a detraction from the main narrative towards engineering a DLA-producing strain.

1. Floss, H. G., et al. (1968). "Biosynthesis of ergot alkaloids. Evidence for two isomerizations in the isoprenoid moiety during the formation of tetracyclic ergolines." Journal of the American Chemical Society **90**(23): 6500-6507.
2. Cheng, J. Z., et al. (2010a). "A role for old yellow enzyme in ergot alkaloid biosynthesis." Journal of the American Chemical Society **132**(6): 1776-1777.
3. Cheng, J. Z., et al. (2010b). "Controlling a structural branch point in ergot alkaloid biosynthesis." Journal of the American Chemical Society **132**(37): 12835-12837.
4. Jakubczyk, D., Caputi, L., Hatsch, A., Nielsen, C. A., Diefenbacher, M., Klein, J., . . . Naesby, M. (2015). Discovery and reconstitution of the cycloclavine biosynthetic pathway—enzymatic formation of a cyclopropyl group. *Angewandte Chemie International Edition*, *54*(17), 5117-5121.
5. Matuschek, M., et al. (2011). "New insights into ergot alkaloid biosynthesis in *Claviceps purpurea*: an agroclavine synthase EasG catalyses, via a non-enzymatic adduct with reduced glutathione, the conversion of chanoclavine-I aldehyde to agroclavine." Organic & biomolecular chemistry **9**(11): 4328-4335.
6. Yan, L. and Y. Liu (2019). "Insights into the mechanism and enantioselectivity in the biosynthesis of ergot alkaloid cycloclavine catalyzed by Aj_EasH from *Aspergillus japonicus*." Inorganic chemistry **58**(20): 13771-13781.

Reviewer #2 (Remarks to the Author):

The authors described the identification of several enzymes for D-lysergic acid biosynthesis from fungi. They engineered yeast strains and introduced the identified enzymes lead to a titer of 71.5 µg/L in shake flask. The work is interesting, but the description of the results in this manuscript can be improved. In the abstract and the introduction part, the authors can describe more about the development of metabolic engineering and synthetic biology strategies used for D-lysergic acid and other alkaloids. These two parts can be improved.

We appreciate the reviewer's helpful comment on the organization and presentation of our manuscript. For brevity in the abstract and introduction, we have revised the abstract and introduction to state our overarching goals and included more details describing the development of our strategy in the section "Biosynthetic resolution of the ergot alkaloid pathway". We have also revised our conclusion to tie up the application of synthetic biology methods in our strain engineering process.

The identification of the enzymes is essential for DLA biosynthesis in engineered yeasts, but the authors described little about enzyme recovery in the manuscript. Even in the methods part, the authors only gave limited information. The authors can strengthen this part in the manuscript.

We thank the reviewer for the insightful comment. We have included more details describing the application of the EFI-EST tools for identifying alternative sequences of our targeted genes to test for their functional expression in yeast in Fig. S6. As the full description detailing the theory, application, and instructions for the use of the EFI-EST has been published (Gerlt, et al. (2015); Gerlt, et al. (2017); Zallot, et. al. (2019)), it would be superfluous to have included an extensive description in the main text.

1. Gerlt, J. A., et al. (2015). "Enzyme function initiative-enzyme similarity tool (EFI-EST): A web tool for generating protein sequence similarity networks." *Biochimica Et Biophysica Acta (BBA)-Proteins and Proteomics* **1854**(8): 1019-1037.
2. Gerlt, J. A. (2017). "Genomic Enzymology: Web Tools for Leveraging Protein Family Sequence-Function Space and Genome Context to Discover Novel Functions." *Biochemistry* **56**, 4293-4308.
3. Zallot, R., Oberg, N. & Gerlt, J.A. (2019). "The EFI Web Resource for Genomic Enzymology Tools: Leveraging Protein, Genome, and Metagenome Databases to Discover Novel Enzymes and Metabolic Pathways." *Biochemistry* **58**, 4169-4182.

The authors can link the supplementary information and the main body of this manuscript, which might help the readers to understand this manuscript. In Fig. 1, the author described the complete biosynthetic pathway of the ergot alkaloids. However, the authors described little about the derivatives. **Fig.1 needs to be focus on the DLA biosynthetic pathway.**

We thank the reviewer for their helpful comment. We have re-designed Fig. 1 to focus on the key pathway to DLA and replaced the representation of the derivatives from the branchpoints with a textual description and put them out of focus.

Line 104 where does this easE_Aj come from? No description of this gene

Line 115-116, easE_Aj was not described in previous content.

We appreciate the reviewer's comment pointing out the gap in our description of easE. We have included in the section "Screening for the functional expression of easE in yeast", a direct reference to the publication that first described the functional heterologous expression of easE_Aj in yeast, as well as more details as to how the other easE sequences were derived.

Fig 2C, the colors of ease_Ec and easE_AI are hard to read.

We thank the reviewer for highlighting this flaw in our figures. We have edited all our figures to use a more legible colour scheme.

The cartoon description of Fig. 2A, Fig.3B, Fig. 4B, Fig. 5A, 5B, 5C, Fig.6A provided little information, they should be moved to the supplementary files. Giving strain details is enough.

We appreciate the reviewer's helpful comment and have removed the cartoon descriptions from the mentioned figures. We have instead described the screening strains used and referenced the supplementary table detailing the genotypes of the strains used.

In the conclusion part, most description in the 1st paragraph (line 236-245) is not related to the biosynthesis of DLA. This should not be in the conclusion part. The same to paragraph 2 of the conclusion part, the authors need to improve this conclusions part.

We appreciate the reviewer's constructive comment and have revised our conclusion to focus on the biosynthesis of DLA before discussing the potential implications of developing an engineered strain for DLA production.

Reviewers' Comments:

Reviewer #1:

Remarks to the Author:

The authors carefully revised the manuscript, which is almost ready for publication. I had only one small suggestion

1. In Fig 3 and else, the enzyme format of easA, easG, and easH should be corrected to EasA, EasG, and EasH. No italic with enzyme, but italic format on gene name. Thus the promoter PTEF2, PGPM1, PGAL10 were all not correct. TEF2, GPM1 and GAL10 should be lowercase and italic. Please check the whole manuscript.

Reviewer #2:

Remarks to the Author:

The authors revised the manuscript totally, and some experiments were added in the revised manuscript. However, problems are still available.

For the SSN strategy, in fact, only few successful examples are available, that's why I ask the authors to provide details about this bioinformatic tools. Recovering enzymes from 9000 sequences is not easy. The selection of the right nodes in the SSN is difficult. I can't see too many advantages of SSN over sequence similarity analysis or phylogenetic analysis. Clustering genes with identity might be better than SSN. In all, the author didn't tell too much about SSN strategy. In the revised abstract part, it is hard to understand. Half of the abstract to describe the essential pharmaceutical characters of ergot alkaloids, and the other part jumps to the biosynthesis of D-lysergic acid and the yeast fermentation results. No connection is available in the abstract, which might be difficult for the readers to understand the research highlights and aims.

As authors stated in the introduction part, the microbial fermentation strategy has been used for DLA production at large-scale. The advantages of current study over the available microbial fermentation are not described at the results and discussion part. Moreover, the authors claimed two key limitations of the available production methods. The authors can't demonstrate their lab-scale work is better than current large-scale production strategy. For the 2nd limitation, the degeneration over the cultivation and preservation processes might be easily solved in industry. For the chemical synthesis, the authors did not give a clear answer if the chemical synthesis of DLA is possible in application. Besides, the highlights of their study are not summarized in the revised manuscript.

For the fermentation results, I noticed there are error bars for the data. I want to know if the authors performed the experiments in triplicates, or just test the data in three times. Normally, 4-L fermentation might get higher titer than 1-L fermentation. Another concern is that the authors can evaluate the final yield and rate of the DLA production. I do not believe 2 mg/L titer have the potential to be applied in industry. The authors should perform the fermentation for at least two times with replicates.

The conclusion part is too long, and most of them are the real conclusion. Thus, moving some of the conclusion description to result and discussion part would be better.

In the revised manuscript, the authors used several times of 'To the best of our current knowledge' or 'the first demonstration', this is quite annoying.

Though the authors come from English speaking countries, I still think the language and writing logic in the revised manuscript is hard to follow.

RESPONSE TO REVIEWER COMMENTS

Reviewer comments in Black, our response in Red, quotes from revised manuscript in Blue.

Reviewer #1 (Remarks to the Author):

The authors carefully revised the manuscript, which is almost ready for publication. I had only one small suggestion:

1. In Fig 3 and else, the enzyme format of easA, easG, and easH should be corrected to EasA, EasG, and EasH. No italic with enzyme, but italic format on gene name. Thus the promoter PTEF2, PGPM1, PGAL10 were all not correct. TEF2, GPM1 and GAL10 should be lowercase and italic. Please check the whole manuscript.

We thank the reviewer for pointing out the oversight on our part. We have since checked the manuscript for the naming of genes and enzymes and corrected the naming formats for both genes and enzymes.

Reviewer #2 (Remarks to the Author):

The authors revised the manuscript totally, and some experiments were added in the revised manuscript. However, problems are still available. For the SSN strategy, in fact, only few successful examples are available, that's why I ask the authors to provide details about this bioinformatic tools. Recovering enzymes from 9000 sequences is not easy. The selection of the right nodes in the SSN is difficult. I can't see too many advantages of SSN over sequence similarity analysis or phylogenetic analysis. Clustering genes with identity might be better than SSN. In all, the author didn't tell too much about SSN strategy.

We appreciate the reviewer's insightful comments on the use of the EFI-EST tools. The reviewer has made a strong point for the inclusion of greater details on our application of the EFI-EST for the reconstitution of the DLA pathway; this however would detract from the main narrative of the manuscript. We have instead, included a supplementary discussion as well as included a reference in the main text to the said discussion to better describe in detail, our approach to the use of these tools in our application.

In the revised abstract part, it is hard to understand. Half of the abstract to describe the essential pharmaceutical characters of ergot alkaloids, and the other part jumps to the biosynthesis of D-lysergic acid and the yeast fermentation results. No connection is available in the abstract, which might be difficult for the readers to understand the research highlights and aims.

We thank the reviewer for the helpful comment on our abstract. We have revised it to better bridge the logic between the description of pharmaceutical application of the ergot alkaloids to the relevance of being able to biosynthesize them.

As authors stated in the introduction part, the microbial fermentation strategy has been used for DLA production at large-scale. The advantages of current study over the available microbial fermentation are not described at the results and discussion part. Moreover, the authors claimed two key limitations of the available production methods. The authors can't demonstrate their lab-scale work is better than current large-scale production strategy. For the 2nd limitation, the degeneration over the cultivation and preservation processes might be easily solved in industry. For the chemical synthesis, the authors did not give a clear answer if the chemical synthesis of DLA is possible in application. Besides, the highlights of their study are not summarized in the revised manuscript.

We greatly appreciate the reviewer's insightful comment about linking the outcomes of our work to resolving the existing issues with DLA production. We have revised the manuscript to better highlight the linkage between the problem statement and the application of our work towards resolving this problem, as well as including a few lines in the conclusion to bring up the highlights of our work.

At our current production titres, we are not yet able to resolve the issues with commercial production in its entirety. Our work however, does set the precedence as a preamble to how these issues can be resolved with further strain and bioprocess optimization:

"With the existing parameters, performing a similar fed-batch cultivation at the 1000 L scale would produce 1.5 g per reactor in approximately a week and a half, and in a year, this would produce 52 g of DLA (with extrapolation, a 100-tonne bioreactor facility would produce 5.2 tonnes of DLA annually); such a level of production is still a preamble to something scalable to the 10-15 tonnes required annually. It is however conceivable that with further bioprocessing and strain optimization, commercial level titres could be attainable."

With regard to the degeneration over the cultivation and preservation processes, while the reviewer has astutely pointed out that there are mitigation measures employed by industry, these measures have not completely ameliorated the issue, which is why they persist and are taken into account as a factor raising the existing cost of production. We propose that these degeneration issues could be better dealt with in a host that is amenable to submerged cultivation in the first place, such as yeast. A proper and complete demonstration of this point however, would require a stability test at the industrial scale, which the strain is not ready for at this point in time.

We have included in the manuscript a clear description of the chemical synthesis of DLA and the current lack of use of chemical synthesis of DLA in industry:

"A number of chemical total synthesis routes towards DLA have been reported¹¹. However, such routes are highly complex, requiring 8 to 19 chemical transformation steps, and the product is often not enantiomerically pure. For example, the highest yielding process requires 19 steps and produces a reported yield of 12%¹². In contrast, the simplest method is a 8-step process that produces a reported yield of 10.6% but is not enantioselective¹³. These issues severely impede the use of chemical synthesis to meet the commercial demand for DLA, which is evident in their lack of use in industry."

For the fermentation results, I noticed there are error bars for the data. I want to know if the authors performed the experiments in triplicates, or just test the data in three times. Normally, 4-L fermentation might get higher titer than 1-L fermentation. Another concern is that the authors can evaluate the final yield and rate of the DLA production. I do not believe 2 mg/L titer have the potential to be applied in industry. The authors should perform the fermentation for at least two times with replicates.

We thank the reviewer for the helpful comment. The fermentation samples were quantitated by standard addition, and thus the error bars were derived from the regression line obtained from this method. In essence, the error bars were representative of the technical replicates of the samples across each concentration level measured. Since then, we have performed two additional biological replicates for both 4 and 1 L fermentations and have included these values in the new figures and tables.

From these biological replicates, we have also found that the 4 L titres are not significantly different from the 1 L titres. Typically, as the reviewer has pointed out, it is expected for larger volume fermentations to produce higher titres of production. We believe that we are not currently able to do so, because the cell densities achieved with the larger fermentation volumes were no greater than the 1 L scale fermentations. This suggests we have most probably hit some form of biological limit at the existing bioprocess parameters, which is not entirely surprising considering that little bioprocess optimization has been performed. Further improvements to the fermentation process that circumvent this limit would likely improve biomass accumulation and increase DLA production titres.

With regard to the evaluation of the final yield and rate of DLA production, we have since included additional lines (lines 255 – 260) to discuss how our current achievable titres compare to the annual global demand for DLA. Given the existing production parameters, our strain is not directly applicable for commercial production. It does however, serve as a proof-of-concept and a prototype strain. Additional strain and bioprocess optimization could improve the scale of production to industrially viable levels.

“With the existing parameters, performing a similar fed-batch cultivation at the 1000 L scale would produce 1.5 g per reactor in approximately a week and a half, and in a year, this would produce 52 g of DLA (with extrapolation, a 100-tonne bioreactor facility would produce 5.2 tonnes of DLA annually); such a level of production is still a preamble to something scalable to the 10-15 tonnes required annually. It is however conceivable that with further bioprocessing and strain optimization, commercial level titres could be attainable.”

The conclusion part is too long, and most of them are the real conclusion. Thus, moving some of the conclusion description to result and discussion part would be better.

We appreciate the reviewer’s constructive comment and revised our conclusion to more succinctly sum up the key points of our manuscript. The more descriptive parts of our initial conclusion were also blended into the end of the discussion (lines 261 – 269).

“This work builds on the growing body of work demonstrating the use of industrially tractable microorganisms for production of complex natural products using economical and renewable feedstocks, such as what has been done for the biosynthesis of the opioids²⁹. Engineered strains provide an excellent platform to drive the discovery of semi-synthetic therapeutic lead compounds, as well as for developing pilot strains for producing important naturally derived therapeutics.

Lastly, with the recent renaissance of research into repurposing psychedelic compounds for anti-depressives and anti-anxiolytics applications³⁰, we believe that our strain could be used to support efforts to probe the natural and semi-synthetic chemical space of ergoline-based therapeutics, to identify novel leads with enhanced therapeutic potential and fewer adverse effects.”

In the revised manuscript, the authors used several times of ‘To the best of our current knowledge’ or ‘the first demonstration’, this is quite annoying. Though the authors come from English speaking countries, I still think the language and writing logic in the revised manuscript is hard to follow.

We thank the reviewer for the helpful comment highlighting the areas of our prose that would benefit from better organization. We have since revised our manuscript to omit the use of the mentioned phrases and to streamline the logic of our manuscript for better comprehension.

Reviewers' Comments:

Reviewer #2:

Remarks to the Author:

I am stasfied with this revised version,however, some small mistakes need futher revision.

For example:

In the abstract,

Figure 6: I only could see g/L, then there are black box. The authors need to check carefully.

In the conclusion part, does away with should be replaced with other scientific word.

Besides, I still think the EFI-EST should not be emphasized too much in the manuscript, especially in the conclusion part.

RESPONSE TO REVIEWER COMMENTS

Reviewer comments in Black, our response in Red, quotes from revised manuscript in Blue (tracked).

Reviewer #2 (Remarks to the Author):

I am stasfied with this revised version,however, some small mistakes need futher revision.

For example:

In the abstract,

We greatly appreciate the reviewer's careful and insightful comments. We have edited the abstract for correctness.

Figure 6: I only could see g/L, then there are black box. The authors need to check carefully.

We have edited Figure 6 for correct display of content.

In the conclusion part, does away with should be replaced with other scientific word.

We thank the reviewer for the helpful comment. We have revised it accordingly by replacing the "does away with" term with "eliminates":

An engineered strain to produce the key ergoline-derivative API, DLA, eliminates the need to hydrolyse ergopeptines and minimizes purification and downstream processing complexity.

Besides, I still think the EFI-EST should not be emphasized too much in the manuscript, especially in the conclusion part.

We greatly appreciate the reviewer's insightful comment. We have removed the emphasis on the EFI-EST by revising the conclusion statement accordingly:

In this work, we described the identification of alternative orthologues of the fastidious enzymes along the ergot alkaloid pathway, and along with it the development of customized strains for their systematic screening.